# Diamond-Coated Plasma Probes for Hot and Hazardous Plasmas

**DOI:** 10.3390/ma13204524

**Published:** 2020-10-13

**Authors:** Codrina Ionita, Roman Schrittwieser, Guosheng Xu, Ning Yan, Huiqian Wang, Volker Naulin, Jens Juul Rasmussen, Doris Steinmüller-Nethl

**Affiliations:** 1Institute for Ion Physics and Applied Physics, University of Innsbruck, A-6020 Innsbruck, Austria; codrina.ionita@uibk.ac.at; 2Institute of Plasma Physics, Chinese Academy of Sciences, Hefei 230031, China; gsxu@ipp.ac.cn (G.X.); yanning@ipp.ac.cn (N.Y.); pst-mail@ipp.ac.cn (H.W.); 3Department of Physics, Technical University of Denmark, DK-2800 Kongens Lyngby, Denmark; vona@fysik.dtu.dk (V.N.); jjra@fysik.dtu.dk (J.J.R.); 4CarbonCompetence GmbH, Weisstraße 9, A-6112 Wattens, Austria; doris.steinmueller@carboncompetence.com

**Keywords:** plasma, plasma probes, hot plasma, hazardous plasma, graphite sputtering, re-deposition, ultra-nano-crystalline diamond coating

## Abstract

Plasma probes are simple and inexpensive diagnostic tools for fast measurements of relevant plasma parameters. While in earlier times being employed mainly in relatively cold laboratory plasmas, plasma probes are now routinely used even in toroidal magnetic fusion experiments, albeit only in the edge region, i.e., the so-called scrape-off layer (SOL), where temperature and density of the plasma are lower. To further avoid overheating and other damages, in medium-size tokamak (MST) probes are inserted only momentarily by probe manipulators, with usually no more than a 0.1 s per insertion during an average MST discharge of a few seconds. However, in such hot and high-density plasmas, their usage is limited due to the strong particle fluxes onto the probes and their casing which can damage the probes by sputtering and heating and by possible chemical reactions between plasma particles and the probe material. In an attempt to make probes more resilient against these detrimental effects, we tested two graphite probe heads (i.e., probe casings with probes inserted) coated with a layer of electrically isolating ultra-nano-crystalline diamond (UNCD) in the edge plasma region of the Experimental Advanced Superconducting Tokamak (EAST) in Hefei, People’s Republic of China. The probe heads, equipped with various graphite probe pins, were inserted frequently even into the deep SOL up to a distance of 15 mm inside the last closed flux surface (LCFS) in low- and high-confinement regimes (L-mode and H-mode). Here, we concentrate on results most relevant for the ability to protect the graphite probe casings by UNCD against harmful effects from the plasma. We found that the UNCD coating also prevented almost completely the sputtering of graphite from the probe casings and thereby the subsequent risk of re-deposition on the boron nitride isolations between probe pins and probe casings by a layer of conductive graphite. After numerous insertions into the SOL, first signs of detachment of the UNCD layer were noticed.

## 1. Introduction

Plasma probes are well established diagnostic tools for the determination of various relevant plasma parameters, such as electron and ion density, electron temperature, floating potential and plasma potential. Although usually accredited to Irving Langmuir [1] and often also denominated with his name, William Crookes [2,3] used plasma probes as early as in the late 19th century. Among the above-mentioned parameters, the plasma potential is of paramount importance for the determination of electric fields in plasmas and for particle and energy transport.

Probes are small additional electrodes of a refractory material (usually graphite or tungsten), enclosed in heat-resistant isolating tubes (usually alumina (Al_2_O_3_) or boron nitride (BN)), inserted into plasma and biased with a variable voltage to register their current–voltage (*IV*-) characteristic, or simply their floating potential. Probes are not complicated, but relatively easy to construct and to handle and are inexpensive. They also have good spatial and temporal resolution. An exact theory of probes is very complicated but usually not necessary, since even with simple approximations valuable information can be drawn from a probe’s *IV*- characteristic.

The simplest version is the cold Langmuir probe (CLP), but with such a probe, the plasma potential can only be derived indirectly from the probe’s *IV*- characteristic. Therefore, particular research efforts have been made on various methods of obtaining a reliable *direct* measure of the plasma potential via the probe’s floating potential. To this end, electron-emissive probes (EEP) and electron-screening probes (ESP) have been developed (see, e.g., Ionita et al. [4]).

Being material objects inserted into the plasma, care must be taken to ensure that the perturbation of the plasma by a probe is not too strong. Vice-versa, probes can be damaged in hot and high-density hazardous plasmas since strong particle fluxes onto the probe and its casings can cause sputtering and heating of the material of the probe casings and of the probes themselves.

The plasma in toroidal fusion experiments, such as tokamaks [5] or stellarators [6], is obviously very hot and rather dense, which, in principle, makes the application of probes self-forbidding. Nevertheless, in tokamaks, up to the so-called medium-size tokamaks (MST), probes have successfully been used in the outer regions of the plasma torus on the low field side (LFS), specifically in the SOL (scrape-off layer). There, the radial particle transport, Reynolds stress and radial transport of poloidal momentum have successfully been determined with probes (see, e.g., [7,8,9,10,11,12]). In the SOL, plasma temperature and particle flux dropped from the plasma interior to values where very short insertions of probes are feasible and highly useful.

In tokamaks, assemblies of various probes, mounted in special probe casings (the so-called *probe heads*), are inserted only for 0.1 s at most in these regions and pulled out quickly again to avoid overly strong exposure to the particle flux (see the references above). This is usually performed by means of pneumatic drives. Hence, these probes are called “reciprocating probes”. Unfortunately, in the literature, the word “probe” is ambiguous since sometimes it is used just for one probe (CLP, EEP, ESP or retarding field analyzer); sometimes, however, it is used for the entire probe head, i.e., the casing plus the assembly of various probes.

Probe measurements inside the LCFS (last closed flux surface) have up to now been ventured only rarely because of the rising particle flux a probe is exposed to, the deeper inside the probe is inserted. The LCFS is infinitesimally close to the separatrix between closed poloidal magnetic field lines and open field lines which connect the plasma with material surfaces in the so-called divertor region. It is also the inner boundary of the SOL.

If the probe pins and their casings are of graphite, sputtering and evaporation can not only lead to damages, but the sputtered-off electrically conductive graphite can also deposit on the boron nitride material usually used for electrical isolation of the probe pins and lead to unwanted and possibly dangerous shunts to ground and/or to neighboring probes, or even short circuits.

Three examples of applications of probes in hot and/or hazardous plasma are briefly described in the following to illustrate the possibilities and limits of probes in such plasmas:In the Axial Symmetric Divertor Experiment (ASDEX) Upgrade (one of the European MSTs) at the Max-Planck Institute for Plasma Physics in Garching near Munich [13], a graphite probe head (the so-called Innsbruck-Padua Probe (InnPadP)) with six cylindrical graphite pins [8,9,12] was inserted frequently up to about 4 mm inside the LCFS [14] (see Figure 1a). One of the probe pins (visible in the upper left corner) is protruding by 3 mm from the other five pins to also enable the measurement of radial profiles of the floating probe potential to obtain information on the radial electric field [11,14]. After numerous shots, the probe pins were visibly sputtered in such a way that the tips were “sharpened”, while their length was not altered (see Figure 1b).In a high-power impulse magnetron sputtering discharge (HiPIMS) [15] in copper, a combination of two CLPs and one EEP was used [16,17,18] (see Figure 2). The wires consisted of tungsten enclosed in alumina tubes. Within less than an hour, the Al_2_O_3_ tubes were covered by the sputtered material (Cu) and made electrically conductive. Only the EEP heating wire of tungsten remained clean due to the heating up to sufficient temperatures for electron emission, i.e., about 2000 K. Furthermore, the upper part of the double-bore Al_2_O_3_ tube carrying the tungsten loop of the EEP remained more or less uncoated.In the chemically very reactive magnetized potassium plasma of a Q-machine [19,20], W-wire probes and circular plane W-probes were also used, and the changes of their surfaces and of their *IV*- characteristics due to chemical reaction between potassium and Al_2_O_3_ and carbon hydrates were investigated [21,22]. These probes were special constructions with a heating wire around the Al_2_O_3_ tube which carried the probe wire. An additional BN tube was pushed over the heating wire for protection (see Figure 3). The heating was not as fully effective as intended, and it took several minutes until the W-probe could be considered sufficiently clean.

In the meantime, a probe head [4,23] (called: the “New Probe Head”—NPH) has been developed for measurements in the EUROfusion MSTs (see below), combining two CLPs, an EEP and two retarding field analyzers (RFA) for ion energy measurements. The NPH also carries two magnetic pickup coils (MPC) to measure magnetic field fluctuations on two radial positions [24,25]. The NPH will be robust enough to withstand the strong plasma heat fluxes and particle fluxes in the edge regions of toroidal magnetic fusion experiments and will make it possible to simultaneously measure plasma potential, electron and ion temperature, electron and ion density and magnetic fluctuations. The use of the NPH is envisaged in all three present European MSTs for comparative measurements of transport parameters:ASDEX Upgrade (Axial Symmetric Divertor Experiment—further abbreviated as AUG) at the IPP (Max-Planck Institute for Plasma Physics) in Garching near Munich, Germany [13].TCV (Tokamak à Configuration Variable) at the Swiss Plasma Center (SPC) of the EPFL (École Polytechnique Fédérale de Lausanne) in Lausanne, Switzerland [26].MAST-U (Mega-Ampere Spherical Tokamak) at the CCFE (Culham Centre for Fusion Energy) in Culham, UK [27].

Furthermore, in the world’s largest stellarator, Wendelstein 7-X of the Max-Planck-Institute for Plasma Physics in Greifswald, Germany [28], the application of a type of NPH is envisaged to compare the measurements of transport parameters with those of MSTs.

The probe casings described in the following were provided by the EAST Group (Experimental Advanced Superconducting Tokamak) in Hefei, China [29,30]. They were sent to Innsbruck to be coated by UNCD (ultra-nano-crystalline diamond) and then sent back to Hefei (China), where they were equipped with various probe pins and mounted on the reciprocating probe manipulator with interesting and promising results [31,32].

As this is a journal on materials, we report here on the behavior, the properties and the resilience of the UNCD-coated probe casings when they were used in the still rather hot and dense edge plasma of EAST. In Section 3.3., a few results of edge plasma measurements in EAST with one of the probe heads are described as examples. It is, however, beyond the scope of this report to discuss these results and their relevance for the edge plasma behavior in EAST in detail. To this end, please see, e.g., [29,30].

## 2. Materials and Methods

The two graphite probe casings of EAST had diameters of 3 cm each (see Figure 4) and overall lengths of 10 cm. They were coated with UNCD by the KOMET RHOBEST Company in Innsbruck, Austria (now [33]). The duration of coating in the vacuum chamber was 14 h. The thickness of the UNCD layer on the top of the probe casings was in the range of 10 to 15 μm, extending over the front side of the probe heads and on the side walls down to about 3 cm towards the rear sides. The thickness of the UNCD layer decreased gradually further on the side walls down to the length of the probe casings of 10 cm.

The UNCD coating was performed by the hot-filament chemical vapor deposition (CVD method [34] using tantalum rods instead of tungsten wires for thermal dissociation of hydrogen. Tantalum rods were heated by electric current to approx. 2200–2400 °C, and the precursor gas mixture was 3% methane diluted in hydrogen. H_2_ is dissociated into atomic H, and methane (CH_4_) formed radicals—mainly CH_3_ and CH_2_—by pyrolysis, enhancing their reactivity. The graphite probe casings were placed on a separate holder made of graphite located in a distance of 2 cm from the filaments horizontal above the probe. The casings were heated by radiation from the filaments up to 750 °C.

The reaction of the activated precursor gas radicals causes the deposition of a solid film starting on the seeded diamond nuclei. Before coating, the graphite probe casings were seeded by detonation of diamond nanoparticles (average size 5 nm) dissolved in isopropyl alcohol in an ultrasonic bath. Based on the nucleation density and roughness of the substrate, the growth on the graphite probe casings followed the Volmer–Weber growth. In [35], the growth of the film by coalescence of islands resulting in island growth is described.

General analyses of the UNCD coatings were carried out on UNCD on Si-wafer substrates. The results of these investigations are published, e.g., in [36,37]. Since the diamond crystals are in the nanometer range (average 5 nm, AFM, XRD) before insertion into the plasma, the UNCD coating has the properties described in these papers. The bulk UNCD coating is electrically isolating, and the surface was H-terminated, i.e., the surface is slightly conductive and hydrophobic (see, e.g., [38]).

Before the first insertion of the fully equipped probe heads, the pure graphite and UNCD-coated parts of the surfaces of the two casings were investigated under a SEM (scanning electron microscope) (see Figure 4). The UNCD layer (Figure 4b) appears in form of spheres, since always a certain number of diamond crystals of 5 nm diameter each forms a cluster. The surface roughness appears rather high, but this had no further effect. The typical appearance of an uncoated graphite surface (Figure 4a) is notably different from the UNCD-coated surface and easily distinguishable.

The roughness was higher than for typical UNCD films due to the rough substrate surface of the graphite probe casings and low nucleation density, leading to 3D island growth and thus to spheres consisting of nano diamond crystals with an average size of 5 nm [37,39]).

After coating, the two probe casings were sent back to Hefei to be equipped by the necessary probe pins. One of the probe heads (Figure 5a) carried three CLPs of which one was protruding radially in order to enable the estimation of all radial profiles of the electric field, similar to earlier investigations (see, e.g., [11,14]). The other probe head (Figure 5b) was mainly used as a Mach probe [40] carrying five graphite pins.

## 3. Results

The coated probe heads were used in numerous discharges in EAST in low (L) mode and high (H) mode. The probe heads were inserted up to 30 mm inside the LCFS, which is exceptionally deep, and performed very well. The most important insights concerning the probe surfaces were the following:

### 3.1. Positive Effects

In contrast to pure, uncovered graphite probe casings, the UNCD-coated surfaces (Figure 4b) were almost not sputtered. In earlier discharges with uncoated probe casings, during deep insertions, a very bad side effect was that graphite from the surface of the probe head was sputtered and re-deposited also on the boron nitride isolators of the probe pins, thereby creating unwanted electric shunts and even short circuits. This effect could be almost completely prevented with our UNCD-coated probe casings.

### 3.2. Negative Effects

The UNCD layer partially dropped off, especially on edged spots. This occurred mainly after taking out the probe head when the experiment was finished. Figure 6a shows the five-pin probe head where a piece of the UNCD-coating had fallen off (marked by a red ellipse) after one night of intensive experiments with numerous insertions of the probe head inside the LCFS. The UNCD drop-off occurred at the edge of the probe head, thus at an especially strongly exposed part of the probe casing.

### 3.3. Exemplary Results

Figure 7 shows an example of a measurement with the UNCD-coated three-pin probe head during the EAST shot #40847 in the SOL, very close to the LCFS. This was the first time this probe head was used. The total injected power into the discharge was > 1.5 MW, at plasma density > 0.5 × 10^19^ m^−3^ and SOL electron temperature > 30 eV.

In Figure 7, the temporal evolutions of the H_α_-line (red line), recorded in the SOL, the floating potential *V_fl_* (blue line) and of the spectrum of the floating potential are shown from top down in the time interval of from *t* = 0 s to *t* = 2.680 s. The H_α_-radiation stems from neutral deuterium in the edge region, which is excited by the plasma electrons. Therefore, the H_α_-line intensity is a measure of the plasma density in this region and thereby also for the radial transport and radial loss, respectively, of plasma.

As indicated by the vertical black lines, a transition from the L-mode to the intermediate mode (I-mode) took place at *t* = 2.520 s and then to the H-mode at *t* = 2.564 s. We clearly see the enhanced H_α_-line intensity during the L-mode, followed by strong fluctuations in the I-mode, which is characteristic for the I-mode. After the transition into the H-mode the H_α_-line intensity strongly drops, being an indication of the better confinement in this latter mode. Later on, though, we see an enhanced fluctuation level in the floating potential, *V_fl_*, in particular at 50 kHz. For further information on the plasma physical results of these investigations and their relevance for edge plasma transport in the EAST, see [29,30].

In numerous further experiments with similar parameters as listed above, the other probe head was inserted between 0 to 30 mm inside the separatrix L-mode and H-mode plasmas. During one insertion, the probe head’s dwelling time inside the separatrix was regularly >200 ms, frequently even >500 ms. During a one-night experiment, about 20–30 shots were carried out.

Once a coated probe was used twice in two separated days. Between these two days, the probe head was disassembled for renewing the probe tips. After the SEM test, the UNCD coating partly fell off when the probe head was re-assembled.

## 4. Conclusions and Outlook

In general, the resilience of the UNCD layers on the probe heads was surprisingly high. The above-mentioned roughness of the UNCD layer surface was perhaps not favorable since it could have caused an inhomogeneous heat transfer to the layer with subsequent temperature gradients and thermal stress across it. This could have increased the tendency of slackening in the UNCD layer.

While the probe casings could quite successfully be protected by the UNCD layer, this can hardly be done with the probe pins since the UNCD layer is electrically isolating, but the probe pins have to be susceptible for the electric particle currents. As shown in Figure 1, after many insertions of the InnPadP into the near SOL of ASDEX Upgrade and even slightly beyond, the six probe pins were visibly affected, causing them to become “sharped”. Thus, this is an example of the effects that the particle fluxes have on bare graphite. An additional effect on graphite probe pins was also observed from time to time, namely, that during deep insertions, some of the pins were heated so strongly that they started to become electron emissive [41].

The durability of the UNCD coating was strongly influenced by the fact that the layer was not closed and graphitic parts of the substrates could have been attacked during plasma insertion. Increasing nucleation density and increasing the coating thickness over the entire probe will lead to improvement in durability.

As shown in Figure 8, different morphologies can be produced by changing the methane concentration and substrate temperature in the above mentioned hot-filament CVD process. In the future, such diamond coatings will be applied, and their positive influence on the durability and performance of thickness, sp^3^-hybridized carbon content and crystal size will be investigated.

According to D. Steinmüller-Nethl, the director of the new UNCD company CarbonCompetence [33], in view of their increased experience with the UNCD coating process [34], for future applications of UNCD coatings of probes and other plasma-facing components (PFC), a more homogeneous, thicker and stabler UNCD coating will be achievable (see Figure 8). Our future research efforts will be directed to take advantage of such new coatings.

## Figures and Tables

**Figure 1 materials-13-04524-f001:**
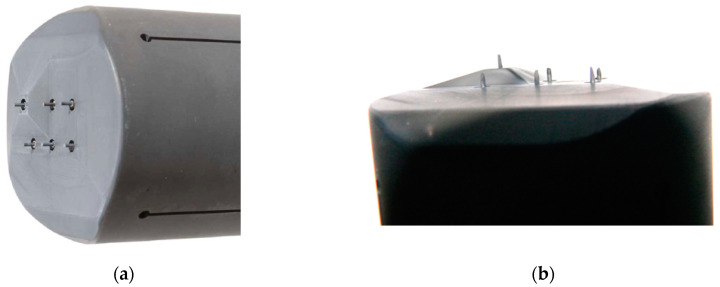
Photo of the Innsbruck-Padua Probe (InnPadP) before (**a**) and after (**b**) numerous insertions into the scrape-off layer (SOL) of the Axial Symmetric Divertor Experiment (ASDEX) Upgrade [8,9,12]. (**a**) The InnPadP graphite casing had a diameter of 50 mm and a length of 115 mm. Its six probe pins (also of graphite) are visible on the left side of the probe casing, each pin consisting of graphite with 1 mm diameter and 3 mm protruding length above the front side of the probe casing. (**b**) The front side of the InnPadP after numerous insertions into the ASDEX Upgrade: the “sharpening” of the probe pins by sputtering is clearly visible, while their length is almost unaltered.

**Figure 2 materials-13-04524-f002:**
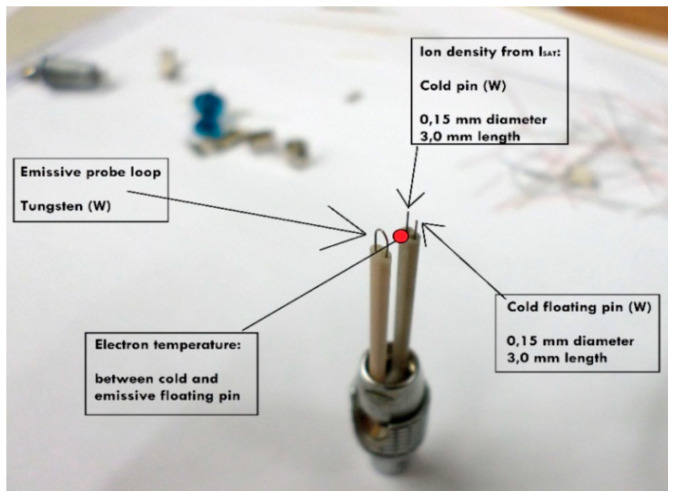
Probe assembly for measurements in a high-power impulse magnetron sputtering discharge (HiPIMS) in Cu, made of W-wire and Al_2_O_3_ tubes, consisting of an electron-emissive probe (EEP), a floating cold Langmuir probe (CLP) and a negatively-biased CLP [16].

**Figure 3 materials-13-04524-f003:**
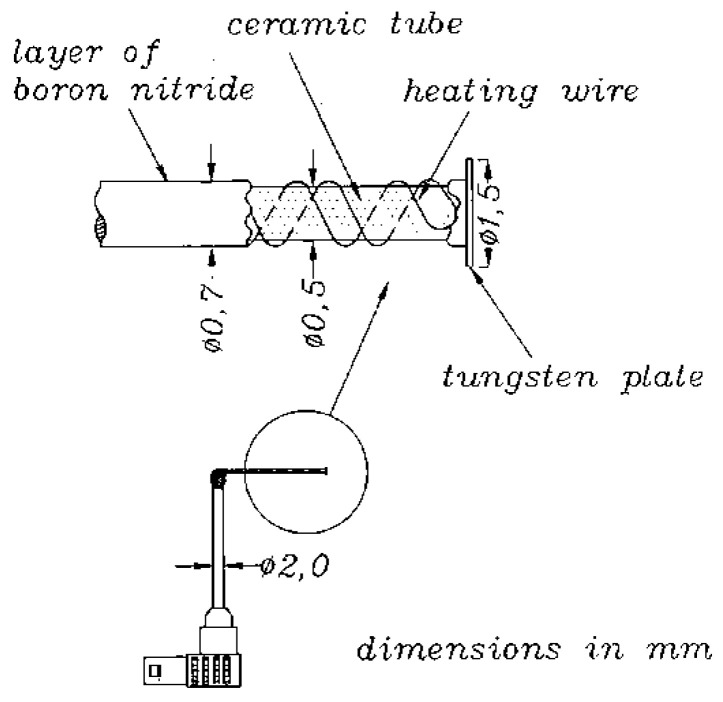
Heatable plane tungsten probe for measurements in a magnetized potassium plasma; drawing taken from [21,22].

**Figure 4 materials-13-04524-f004:**
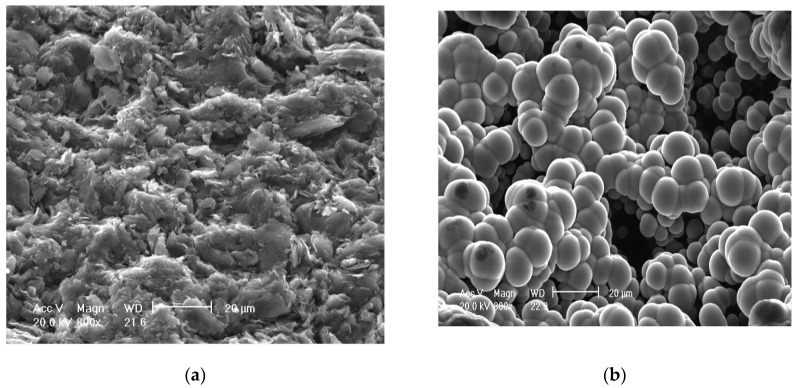
SEM (scanning electron microscope) of the probe surfaces before use: (**a**) uncoated graphite; (**b**) ultra-nano-crystalline diamond (UNCD)-coated surface.

**Figure 5 materials-13-04524-f005:**
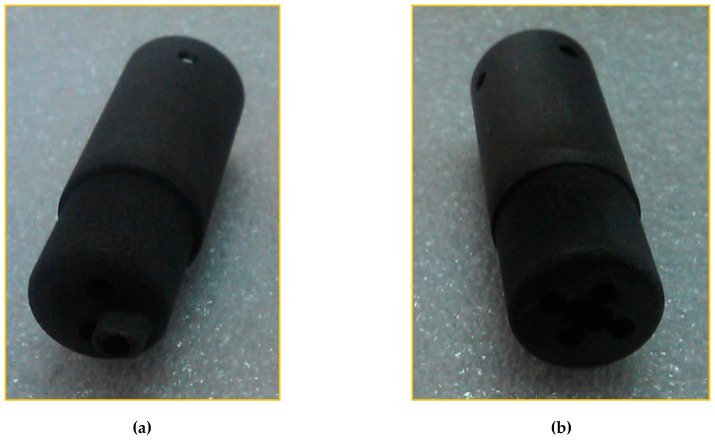
The two graphite probe casings from the Experimental Advanced Superconducting Tokamak (EAST) Group, each with a diameter of 3 cm and overall lengths of 10 cm, shown here without the probe pins inserted, i.e., as they were introduced into the UNCD coating device in Innsbruck: (**a**) the probe casing for three Langmuir probe pins); (**b**) the probe casing for the 5-pin Mach probe.

**Figure 6 materials-13-04524-f006:**
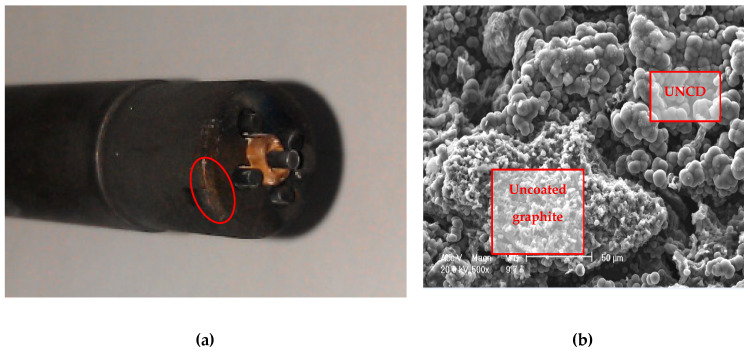
(**a**) Photo of the five-pin probe head after numerous insertions into the EAST SOL and beyond. The red ellipse indicates the spot where the UNCD-coating had dropped off after handling the probe head after taking it out. (**b**) SEM of the spot marked in (**a**). The difference between the still UNCD-coated part and the uncoated is clearly distinguishable.

**Figure 7 materials-13-04524-f007:**
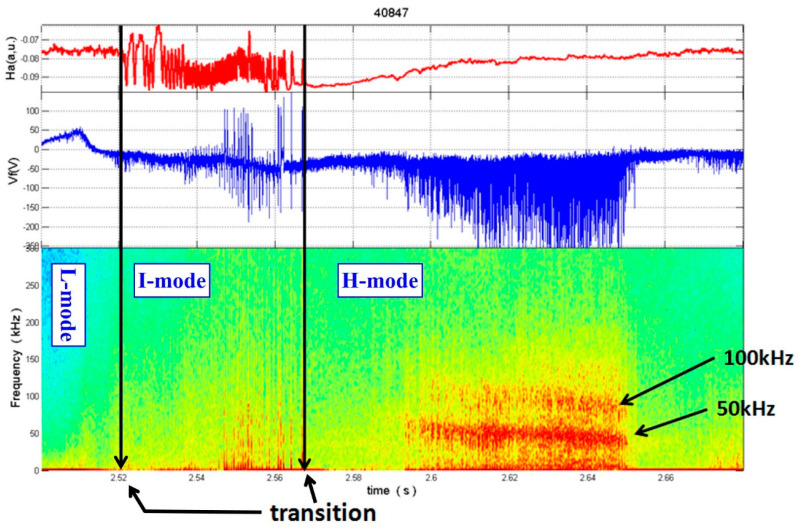
From top to bottom: Temporal evolutions of the H_α_ line intensity (red line) and the floating potential (blue line) measured with the three-pin probe head very close to the LCFS, and of the spectrum of floating potential fluctuations. The two black vertical arrows show the transitions from L-mode (low mode) to I-mode (intermediate mode) and from I- to H-mode (high mode), respectivly.

**Figure 8 materials-13-04524-f008:**
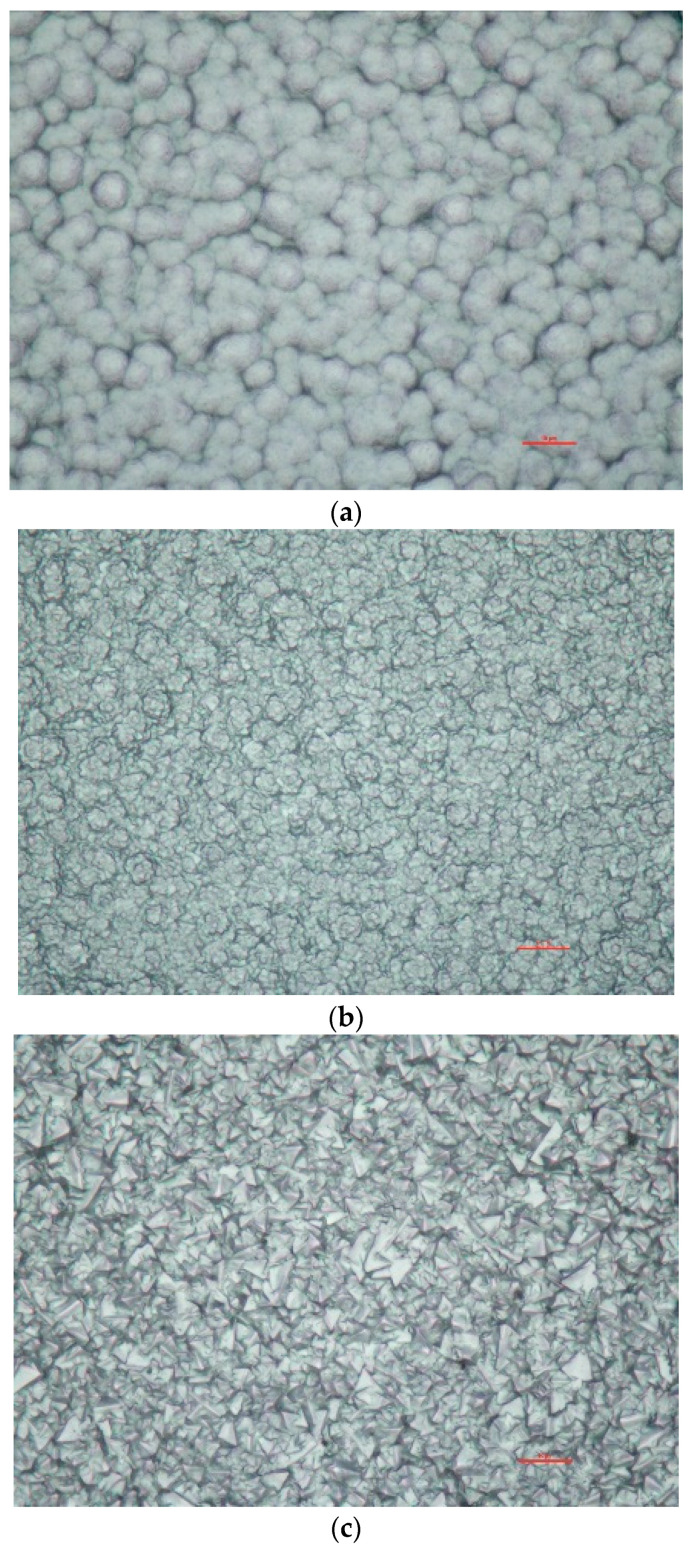
Examples of recent UNCD coatings by the company CarbonCompetence (total size of the samples: 124 × 89 μm): (**a**) nanocrystalline diamond; (**b**) microcrystalline diamond (small grains); (**c**) microcrystalline diamond (medium-sized grains).

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
