# Peer review of "Diamond-Coated Plasma Probes for Hot and Hazardous Plasmas"

_materials, 2020, doi:10.3390/ma13204524_

Round 1
Reviewer 1 Report
In the reviewed manuscript is studied the effect of diamond coating (UNCD) deposited onto a casings of probes intended to be used for the characterization of Tokamak plasma. It is shown that produced diamond coating prolongs the life-time of the probes and allows for the determination of plasma parameters in the edge plasma region. Although the results are interesting, in my opinion the manuscript is not suitable for the publication in the journal Materials as there is no characterization of the studied material and thus the article is more appropriate for plasma-related journal, e.g. MDPI Plasma, or its major revision is mandatory. My main comments are as follows:
- I miss detail description of the deposition procedure/set-up. It is only stated that “special thermal activation of the reactant gas” was used.
- I miss physico-chemical characterisation of the produced coatings – preferably before and after the insertion into the plasma. In fact, only 1 SEM image is shown and discussed in a qualitative way only. What is the chemical structure, electrical properties, roughness ...of the UNCD coating? It is stated that the UNCD layer appears in the form of spheres (size?) composed of 5nm diamond crystals – how the size of the crystals was evaluated is not clear. This information has to be added.
- Authors describe in detail competing probes, but there is no description of the probe developed by them (i.e. the probe assembly).
- Figure 4 shows the probe heads, but in fact it is difficult to see anything on this figure. Also the quality of Fig 6a should be improved.
- Authors state that they have used either 3 or 5-pin probes, but only the results for the 3-pin probe are shown. In contrast, partial detachment of the coating is shown for 5-pin probe. This is confusing.
- It is hypnotized that the durability of the probe may be enhanced when smoother coating will be used. Authors state that one of the company is able to produce such films. But again, no details are provided concerning the deposition procedure. This is not appropriate.
- It should be interesting to compare probes with and without UNCD coating.
Author Response
Please see the attachment, thanks!

Reviewer 2 Report
The paper presents an ultra nano-crystalline diamond (UNCD) coating for plasma probes to measure important plasma parameters. Application examples for different plasma probes are provided, which is of interest. The main issue with the article, however, is that it is not entirely clear which community should benefit from the work (besides probably offering the coating for commercial purposes).
To address readers interested in materials, more information on the UNCD coating process and the coating characteristics should be provided. To address readership in the Tokamak community, the measured parameters should be discussed in more detail.
Figure 1: The six probe pins (also of graphite) are visible on the left right side of the probe 91 casing, …
You might briefly comment on the kind of probes used and why one pin is elevated?
Was the UNCD coating performed by a hot-filament chemical vapour deposition process as mentioned in Ref. [34]? Please give further details. Which part of the probe casing was coated (only front and edges, i.e. deposition from one side)? What might be the substrate temperature during deposition? How is the pronounced particle structure of the coating generated?
Please explain the behavior of the measured floating potential that initially increases to +50 V (what does this mean?) and then drops to negative values. How is floating potential defined in this process? In the introduction plasma potential is mentioned to be of highest relevance. How si plasma potential related to floating potential in the discussed process?
Author Response
Please see the attachment, thanks!

Round 2
Reviewer 2 Report
The revised version substantially improves the readability and the interest to the readers of the manuscript. The added content is thus valuable and justifies publication of the paper.